# Learning Superpoint Graph Cut for 3D Instance Segmentation

**Le Hui[†], Linghua Tang[†], Yaqi Shen, Jin Xie[∗], Jian Yang[∗]**
PCA Lab, Nanjing University of Science and Technology, China
`{le.hui, tanglinghua, syq, csjxie, csjyang}@njust.edu.cn`

## Abstract

3D instance segmentation is a challenging task due to the complex local geometric structures of objects in point clouds. In this paper, we propose a learning-based superpoint graph cut method that explicitly learns the local geometric structures of the point cloud for 3D instance segmentation. Specifically, we first oversegment the raw point clouds into superpoints and construct the superpoint graph. Then, we propose an edge score prediction network to predict the edge scores of the superpoint graph, where the similarity vectors of two adjacent nodes learned through cross-graph attention in the coordinate and feature spaces are used for regressing edge scores. By forcing two adjacent nodes of the same instance to be close to the instance center in the coordinate and feature spaces, we formulate a geometry-aware edge loss to train the edge score prediction network. Finally, we develop a superpoint graph cut network that employs the learned edge scores and the predicted semantic classes of nodes to generate instances, where bilateral graph attention is proposed to extract discriminative features on both the coordinate and feature spaces for predicting semantic labels and scores of instances. Extensive experiments on two challenging datasets, ScanNet v2 and S3DIS, show that our method achieves new state-of-the-art performance on 3D instance segmentation. Code is available at `https://github.com/fpthink/GraphCut`.

## 1 Introduction

In recent years, with the development of 3D sensors, such as LiDAR and Kinect camera, various 3D computer vision tasks have been receiving more and more attention. 3D instance segmentation is a fundamental task in 3D scene understanding and has been widely used in kinds of applications such as self-driving cars, virtual reality, and robotic navigation. Although recent progress in 3D instance segmentation is encouraging, it is still a challenging task due to irregularities and context uncertainties of 3D points in 3D scenes with complex geometric structures.

Many efforts have been dedicated to 3D instance segmentation and achieved promising performance. These methods can be mainly divided into two categories: detection-based methods [44, 45] and clustering-based methods [40, 18]. Among detection-based methods, 3D-BoNet [44] first detects the 3D bounding boxes and then employs a mask prediction network to predict the object mask for 3D instance segmentation. However, for objects with complex geometric structures, detection-based methods [45] cannot obtain accurate 3D bounding boxes, thereby degrading the instance segmentation performance. The clustering-based method SGPN [40] clusters 3D points based

---

[†]Equal Contributions, [∗]Corresponding authors.

Le Hui, Linghua Tang, Yaqi Shen, Jin Xie, and Jian Yang are with PCA Lab, Key Lab of Intelligent Perception and Systems for High-Dimensional Information of Ministry of Education, and Jiangsu Key Lab of Image and Video Understanding for Social Security, School of Computer Science and Engineering, Nanjing University of Science and Technology, China.

36th Conference on Neural Information Processing Systems (NeurIPS 2022).

on semantic segmentation to generate instances. Unlike SGPN, Jiang *et al.* [18] developed an offset branch to cluster points based on semantic predictions in dual coordinate spaces, including original and shifted coordinate spaces. Besides, some follow-up methods utilize tree structures [25], hierarchical aggregation [3], and soft semantic segmentation [37] to boost the performance of 3D instance segmentation. However, most of these clustering-based methods rely on center offsets and semantics to segment instances, which cannot effectively capture the geometric context information of point clouds. Therefore, the performance of instance segmentation is usually limited by objects with complex geometric structures in point clouds.

In this paper, we propose a learning-based superpoint graph cut method that explicitly learns the local geometric structures of point clouds to segment 3D instances. Specifically, we construct the superpoint graph to learn the geometric context similarities of superpoints and convert the instance segmentation into a binary classification of edges. Our method consists of an edge score prediction network to predict edge scores and a superpoint graph cut network to generate instances. In our method, we oversegment the raw point clouds into superpoints and construct the superpoint graph by linking the $k$-nearest superpoints in the coordinate space. In the edge score prediction network, we first perform cross-graph attention on the local neighborhoods of two adjacent nodes to extract local geometric features for measuring the similarity of the nodes. Then, based on the learned similarity vectors from the coordinate and feature spaces, we adopt an edge score branch to predict the edge scores. In addition, we propose a geometry-aware edge loss to train the edge score prediction network by forcing the adjacent nodes of the same instance to be close to the instance center in both the coordinate and feature spaces. In the superpoint graph cut network, we use the learned edge scores combined with semantic classes of the nodes to cut the edges for forming object proposals. The proposals are obtained by applying the breadth-first-search algorithm on the superpoint graph to aggregate nodes in the same connected component. In each proposal, we apply bilateral graph attention to aggregate local geometric features to extract discriminative features for predicting classes and scores of proposals. Furthermore, we adopt a mask learning branch to filter the low-confidence superpoints within the proposal to generate instance.

In summary, we present an edge score prediction network that learns the local geometric features of adjacent nodes for producing edge scores. To train it, we propose a geometry-aware edge loss to keep the instance compact in the coordinate and feature space simultaneously. We present a superpoint graph cut network that extracts discriminative instance features to generate accurate instances by using bilateral graph attention in the coordinate and feature spaces. Extensive experiments on the ScanNet v2 [7] and S3DIS [1] datasets show that our method achieves new state-of-the-art performance on 3D instance segmentation. On the online test set of ScanNet v2, our method achieves the performance of 55.2% in terms of mAP, which is 4.6% higher than the current best results [25]. For S3DIS, our method outperforms the current best results [37] over 2% in terms of mAP.

## 2 Related Work

**3D semantic segmentation.** Extracting features from irregular 3D point clouds is crucial for 3D semantic segmentation. Qi *et al.* [30] first proposed PointNet to learn point-wise features from a point set for semantic segmentation through the multi-layer perceptron network. Following it, many efforts [24, 34, 46, 15, 47, 4, 2] have been proposed to improve semantic segmentation performance. Early point-based methods [36, 43, 31, 14] design various local feature aggregation strategies to extract discriminative point-wise features for semantic segmentation. Inspired by successful 2D convolution networks, view-based methods [23, 35, 17, 19] project the point cloud into multiple regular 2D views, where the regular 2D convolution is applied to extract features. In addition to view-based methods, volumetric-based methods [27, 39, 10, 6] first voxelize the point cloud into regular 3D grids and then apply 3D convolution to extract local features of point clouds. In order to capture the local geometric structures of point clouds, graph-based methods [42, 22, 38, 5, 16] construct the graph on point clouds and utilize graph convolution to aggregate local geometric information for semantic segmentation.

**3D instance segmentation.** 3D instance segmentation is a more challenging task, which further needs to identify each instance. Current methods can be roughly grouped into two classes: detection-based methods and clustering-based methods.

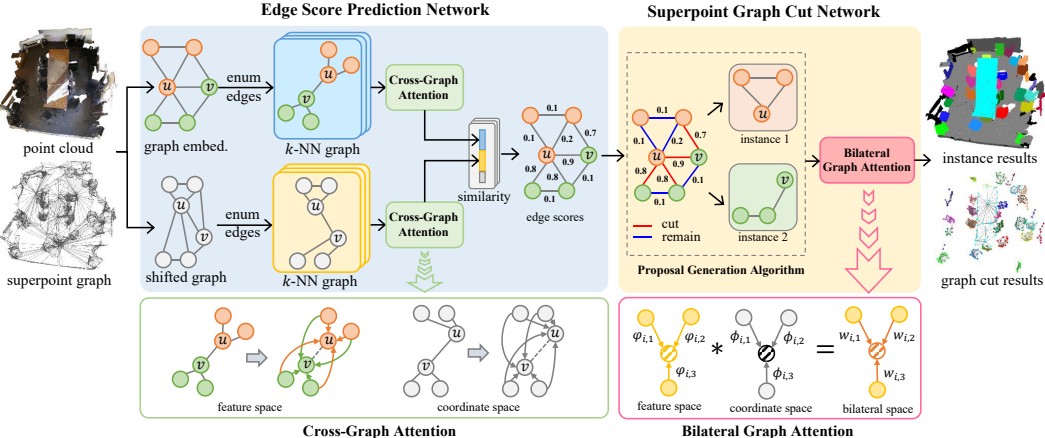

Figure 1: The framework of our learning-based superpoint graph cut method. Given the superpoint graph, we first extract superpoint features (omitted in the figure). Then, we construct an edge score prediction network to extract edge embeddings from the coordinate and feature spaces for predicting edge scores. Finally, based on the learned edge scores, we develop a superpoint graph cut network to generate accurate instances.

Detection-based methods [45, 13, 26] first detect 3D bounding boxes of each object in point clouds, and then apply a mask prediction network on each box to predict the object mask for 3D instance segmentation. In [44], a 3D instance segmentation framework dubbed 3D-BoNet is proposed, which directly regresses the 3D bounding boxes for all instances and predicts point-level masks for each instance. Yi *et al.* [45] proposed a generative shape proposal network that generates proposals by reconstructing shapes from noisy observations in a scene for 3D instance segmentation. In addition, using both geometry and RGB inputs, [13] develops a joint 2D-3D feature learning network that combines the 2D and 3D features to regress 3D object bounding boxes and predict instance masks.

Clustering-based methods usually use point similarity [40], semantic maps [11, 20], or geometric shifts [18, 3, 25, 33] to cluster 3D points into object instances. A similarity group proposal network was proposed in [40] to cluster points by learning point-wise similarity for generating instances. [29] proposes a multi-task learning framework that simultaneously learns semantic classes and high-dimensional embeddings of 3D points to cluster the points into object instances. In [41], a segmentation framework is introduced to learn semantic-aware point-wise instance embedding for associatively segmenting instances and semantics of point clouds. Han *et al.* [11] proposed an occupancy-aware method to predict the number of occupied voxels for each instance. PointGroup [18] clusters points by using predicted point-wise center offset vectors and point-wise semantic labels. The follow-up method [3] adopts a hierarchical aggregation strategy for 3D instance segmentation, which first performs point aggregation to cluster points into preliminarily sets and then performs set aggregation to cluster sets into instances. Lately, Vu *et al.* [37] proposed a soft grouping strategy to mitigate the problem of semantic prediction errors by associating each point with multiple classes, yielding in significant performance gains in 3D instance segmentation. In addition, a semantic superpoint tree network, called SSTNet, is proposed in [25] for segmenting point clouds in instances. It first groups superpoints with similar semantic features to build a binary tree and then generates instances by tree traversal and splitting. To make the network more efficient, a dynamic convolution network combined with a small Transformer network is constructed to propose a lightweight 3D instance segmentation method [12].

## 3 Method

An overview of our learning-based superpoint graph cut method is illustrated in Figure 1. Based on the superpoint graph, the edge score prediction network (Sec. 3.1) extracts edge embeddings from the coordinate and feature spaces for predicting edge scores. After that, the superpoint graph cut network (Sec. 3.2) generates accurate object instances by learning discriminative instance features to

predict classes and scores of instances. Finally, in Sec. 3.3, we describe how to train our method and inference instances from point clouds.

## 3.1 Edge Score Prediction Network

Given a raw point cloud, we oversegment it into superpoints and construct the superpoint graph $\mathcal{G} = (V, E)$, where $V$ represents the node set of superpoints and $E$ represents the edge set. Since the superpoint representation is coarser than the point representation, learning features directly from the superpoint representation cannot effectively capture the local geometric structures of point clouds. Therefore, we apply submanifold sparse convolution [10] on the point cloud to extract point-level features and use the point-level features to initialize superpoint-level features by average pooling. After that, we apply edge-conditioned convolution [32] to extract superpoint features, denoted as $\boldsymbol{F} \in \mathbb{R}^{|V| \times C}$, where $C$ is the feature dimension.

### 3.1.1 Edge Feature Embedding

Once we obtain superpoint features, the edge score prediction network learns edge embeddings to predict edge scores for segmenting instances. Given the adjacent nodes $(u, v) \in E$, it is desired that the learned edge embedding can effectively identify whether nodes $u$ and $v$ belong to the same instance. To tackle this problem, we applies cross-graph attention to the superpoint graph in double spaces (the coordinate and feature spaces) for learning superpoint similarities. The learned similarity vectors of nodes $u$ and $v$ are used to form the edge embedding for predicting edge scores.

**Edge embedding in coordinate space.** To characterize the similarity of nodes $u$ and $v$, we first shift them toward the corresponding instance centroids in the coordinate space. Here, a multi-layer preceptron (MLP) network encodes $\boldsymbol{F}$ to produce $|V|$ offset vectors $\boldsymbol{O} = \{\boldsymbol{o}_1, \dots, \boldsymbol{o}_{|V|}\} \in \mathbb{R}^{|V| \times 3}$. Give the original superpoint coordinates $X = \{x_1, \dots, x_{|V|}\} \in \mathbb{R}^{|V| \times 3}$, the shifted superpoint coordinates $\hat{X} = \{\hat{x}_1, \dots, \hat{x}_{|V|}\}$ are obtained by $\hat{X} = X + O$. In this way, the geometric distance of nodes belonging to different instances will be increased, so that the discrimination of the superpoints will be enhanced. After that, based on the shifted coordinate space, for node $u$, we leverage its $k$-nearest superpoints (i.e., $\mathcal{N}_u$) to construct the local $k$-NN graph $G_u$. Similarly, we can obtain the graph $G_v$ for node $v$. Then, we perform cross-graph attention across $G_u$ and $G_v$ to characterize the similarity of nodes through the learned feature vectors, as shown in Figure 1. Taking the node $u$ as an example, the weight $\boldsymbol{\alpha}$ of cross-graph attention is defined as:

$$\alpha_{u,i} = \text{MLP}(\hat{x}_i - \hat{x}_u), \forall i \in \mathcal{N}_u \cup \mathcal{N}_v \tag{1}$$

where $\hat{x}_i$ and $\hat{x}_u$ are the shifted coordinates. Note that $i$ enumerates the all $2*k$ neighbors across two graphs. Therefore, the final output feature vector can be formulated as:

$$\boldsymbol{h}_u = \sum_{i \in \mathcal{N}_u \cup \mathcal{N}_v} \hat{\alpha}_{u,i} * \text{MLP}(\hat{x}_i) + b_i \tag{2}$$

where $\hat{\alpha}_{u,i}$ is the weight $\alpha_{u,i}$ after softmax and $b_i$ is a learnable bias. The learned feature vector $\boldsymbol{h}_u \in \mathbb{R}^C$ can characterize the geometry similarity by adaptively learning the geometric differences on two graphs. In the same way, we can obtain feature vector $\boldsymbol{h}_v$ for another node $v$. We combine the feature vectors $\boldsymbol{h}_u$ and $\boldsymbol{h}_v$ as the edge embedding in the coordinate space: $\boldsymbol{e}_{u,v} = [\boldsymbol{h}_u, \boldsymbol{h}_v]$.

**Edge embedding in feature space.** In addition to the coordinate space, we also consider the feature space to extract discriminative edge embeddings. Firstly, a MLP network encodes $\boldsymbol{F}$ to produce the initial feature embedding $\boldsymbol{Z} \in \mathbb{R}^D$. By pushing feature embeddings of instances away from each other, we enlarge the gap between different instances in the feature space. Given a pair of nodes $(u, v) \in E$, we construct the $k$-NN graphs $\hat{G}_u$ and $\hat{G}_v$ in the feature space, respectively. In this way, it is expected that each graph can aggregate superpoints within the same instance. Then, we execute cross-graph attention across $\hat{G}_u$ and $\hat{G}_v$ to characterize the similarity of nodes in the feature space. Finally, we can obtain the feature vectors $\hat{\boldsymbol{h}}_u \in \mathbb{R}^C$ and $\hat{\boldsymbol{h}}_v \in \mathbb{R}^C$. If $u$ and $v$ belong to the same instance, they share the similar $k$-NN graph in the feature space, so that the learned feature vectors $\hat{\boldsymbol{h}}_u$ and $\hat{\boldsymbol{h}}_v$ are similar to each other. Here, we also combine the feature vectors to obtain the edge embedding in the feature space: $\hat{\boldsymbol{e}}_{u,v} = [\hat{\boldsymbol{h}}_u, \hat{\boldsymbol{h}}_v]$.

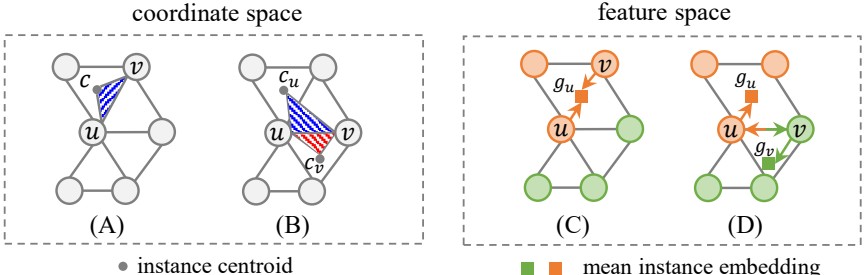

coordinate space       feature space

(A)      (B)      (C)      (D)

● instance centroid      ■ ■ mean instance embedding

Figure 2: The details of our geometry-aware edge loss. $u$ and $v$ in graphs (A) and (C) belong to the same instance, and $u$ and $v$ in graphs (B) and (D) belong to different instances.

**Edge score prediction.** After obtaining edge embeddings in the coordinate and feature spaces, we utilize a simple MLP network to generate the edge score, which is defined as:

$$a_{u,v} = \sigma(\text{MLP}([\boldsymbol{e}_{u,v}, \hat{\boldsymbol{e}}_{u,v}, d_{u,v}])) \tag{3}$$

where $[\cdot, \cdot, \cdot]$ indicates the concatenation operation, $\sigma$ denotes the sigmoid function, and $d_{u,v}$ represents the geometric distance between the nodes $u$ and $v$ in the shifted coordinate space. In the experiment, if the edge score $a_{u,v} > 0.5$, it means that the edge between nodes $u$ and $v$ should be cut from the superpoint graph. We use the binary cross-entropy loss $\mathcal{L}_{edge}$ to minimize the edge score.

### 3.1.2 Geometry-Aware Edge Loss

To train the edge score prediction network, we employ the geometric structures of the superpoint graph to form a geometry-aware edge loss, as shown in Figure 2.

Specifically, given the nodes $u$, $v$ and their corresponding instance centroids $c_u$ and $c_v$, we draw nodes toward their instance centroids by minimizing the $L_2$ distance $d_{u,c_u}$ and $d_{v,c_v}$. Furthermore, when $(u, v) \in E$ belong to the same instance, it is expected that they can collaboratively shift to the same instance centroid $c$ by minimizing the area of triangle $\triangle uvc$. While $(u, v) \in E$ belong to different instances, it is expected that they can collaboratively shift to their own instance centroids $c_u$ and $c_v$ by minimizing the area of triangles $\triangle uvc_u$ and $\triangle uvc_v$. The area constraint in the coordinate space is written as:

$$\mathcal{L}_{area} = \frac{1}{|E|} \sum_{(u,v)\in E} \|\hat{x}_u - c_u\|_2 + \|\hat{x}_v - c_v\|_2 + \frac{1}{2}(|(c - \hat{x}_u) \times (c - \hat{x}_v)|\mathbb{I}(u,v) \tag{4}$$
$$+ (|(c_u - \hat{x}_u) \times (\hat{x}_v - \hat{x}_u)| + |(c_v - \hat{x}_v) \times (\hat{x}_u - \hat{x}_v)|)(1 - \mathbb{I}(u,v)))$$

where $\mathbb{I}(u,v)$ is the indicator function, and $\mathbb{I}(u,v)$ equals to 1 if $u$ and $v$ belong to the same instance, and 0, otherwise. Note that "$\times$" represents the outer product of vectors for computing the area of triangles. For nodes $u$ and $v$ from the same instance, $u$ and $v$ are simultaneously close to the common instance centroid. Therefore, they are pulled close to each other in the coordinate space, which is helpful to group $u$ and $v$ into the same instance. For nodes $u$ and $v$ from different instances, $u$ and $v$ are respectively close to the corresponding instance centroids. Thus, they are pushed away in the coordinate space, which is helpful to divide $u$ and $v$ into two different instances.

Likewise, we expect the nodes in the same instance to be compact in the feature space by constraining their feature embeddings. For $(u, v) \in E$ belong to the same instance, we draw embeddings of $u$ and $v$ toward the mean embedding of the instance, and also pull them to each other. For $(u, v) \in E$ belong to different instances, we push embeddings of $u$ and $v$ away from each other. In addition, the instances are pushed away from each other by increasing the distance of their own mean embedding of instances. Thus, the constraint in the feature space is written as:

$$\mathcal{L}_{feat} = \frac{1}{|E|} \sum_{(u,v)\in E} ([\|\boldsymbol{z}_u - \boldsymbol{z}_v\|_2 - \delta]_+^2 + [\|\boldsymbol{z}_u - \boldsymbol{g}_u\|_2 - \delta]_+^2)\mathbb{I}(u,v) \tag{5}$$
$$+ ([2\beta - \|\boldsymbol{z}_u - \boldsymbol{z}_v\|_2]_+^2 + [2\beta - \|\boldsymbol{g}_u - \boldsymbol{g}_v\|_2]_+^2)(1 - \mathbb{I}(u,v))$$

where $\boldsymbol{z}_u \in \mathbb{R}^D$ and $\boldsymbol{z}_v \in \mathbb{R}^D$ are the feature embeddings. Note that $\boldsymbol{g}_u \in \mathbb{R}^D$ and $\boldsymbol{g}_v \in \mathbb{R}^D$ indicate the mean feature embeddings of the instances that $u$ and $v$ belong to, respectively. The

---

**Algorithm 1** Proposal Generation Algorithm

---

**Input**: node semantic scores $\boldsymbol{S} = \{\boldsymbol{s}_1, \ldots, \boldsymbol{s}_{|V|} \mid \boldsymbol{s}_i \in \mathbb{R}^N \text{ for } i = 1, \ldots, |V|\}$, $N$ is the number of classes; semantic threshold $\theta$; edge scores $\mathbf{A} = \{a_{u,v}\} \in \mathbb{R}^{|E| \times 1}$, $a_{u,v}$ indicates the score of edge which connects nodes $u$ and $v$;

**Output**: proposals $\mathbf{I} = \{I_1, \ldots, I_m\}$, $m$ is the number of proposals.

1: initialize an empty instance set $\mathbf{I}$
2: **for** $i = 1$ to $N$ **do**
3:    **if** $i$ is valid class (excluding wall, floor) **then**
4:       initialize an array $f$ (visited) of length $|V|$ with all zeros
5:       **for** $v = 1$ to $|V|$ **do**
6:         **if** $f_v == 0$ and $s_v^i > \theta$ **then**
7:           initialize an empty queue $Q$
8:           initialize an empty set $I$
9:           $f_v = 1$ ; $Q$.pushBack($v$) ; add $v$ to $I$
10:           **while** $Q$ is not empty **do**
11:             $h = Q$.popFront()
12:             **for** each $k \in \{k \mid a_{h,k} < 0.5\}$ **do**
13:               **if** $f_k == 0$ and $s_k^i > \theta$ **then**
14:                 $f_k = 1$ ; $Q$.pushBack($k$) ; add $k$ to $I$
15:           add $I$ to $\mathbf{I}$
16: **return** $\mathbf{I}$

---

thresholds $\delta$ and $\beta$ are set to be 0.1 and 1.5 to ensure that the inter-instance distance is higher than the intra-instance. Finally, the geometry-aware edge loss is defined as:

$$\mathcal{L}_{geo} = \mathcal{L}_{area} + \mathcal{L}_{feat} + \mathcal{L}_{edge} \tag{6}$$

### 3.2 Superpoint Graph Cut Network

#### 3.2.1 Proposal Generation via Superpoint Graph Cut

Given the edge score $\boldsymbol{A} = \{a_{u,v}\} \in \mathbb{R}^{|E| \times 1}$, we propose a proposal generation algorithm to generate candidate proposals by simultaneously employing the learned edge scores and the predicted semantic classes of nodes (*i.e.*, superpoints). Specifically, in order to mitigate semantic prediction errors, we follow [37] and adopt a soft threshold $\theta$ to associate the nodes with multiple classes. Given semantic scores of superpoints $\boldsymbol{S} = \{\boldsymbol{s}_1, \ldots, \boldsymbol{s}_{|V|} \mid \boldsymbol{s}_i \in \mathbb{R}^N \text{ for } i = 1, \ldots, |V|\}$, where $N$ is the number of classes, if $s_v^i > \theta$, the $v$-th superpoint can be associated with the $i$-th class. In this way, for the $i$-th class, we can slice a superpoint subset $C_i$ on the superpoint graph, where the semantic score of the superpoint on the $i$-th class index is higher than $\theta$. Then, on the superpoint graph, for the edge $(u, v) \in E$, if nodes $u \in C_i$ and $v \in C_i$, the edge $(u, v)$ will be preserved, otherwise the edge will be deleted. In other words, we remove the edge between two superpoint nodes with different semantics. After that, for the preserved edges $(u, v)$ on the superpoint graph, we utilize the edge score $a_{u,v}$ to determine whether the edge should be cut from the superpoint graph. In the experiment, the threshold for cutting the edge is set to 0.5. If the edge score is higher than 0.5, the edge will be cut from the superpoint graph. Finally, we apply the breadth-first-search algorithm on the superpoint graph to aggregate nodes in the same connected component for generating proposals for the $i$-th class. In this way, we can generate proposals for $N$ classes by iterating through $N$ classes. The details are shown in Algorithm 1.

#### 3.2.2 Bilateral Graph Attention for Proposal Embedding

As we obtain proposals $\boldsymbol{I} = \{I_1, \ldots, I_m\}$ from the point cloud, we propose bilateral graph attention to extract proposal embeddings for generating instances by applying the attention mechanism in both the coordinate and feature spaces. Specifically, given the $i$-th proposal, we first compute proposal centroid $c_i$ by averaging the shifted superpoint coordinates. Then, we adopt the inverse distance weighted average of the corresponding superpoints to interpolate the embedding of the proposal

centroid, which is formulated as:

$$\boldsymbol{f}'_i(c_i) = \frac{\sum_{j \in I_i} \psi_j(c_i) * \boldsymbol{f}_j}{\sum_{j \in I_i} \psi_j(c_i)}, \psi_j(c_i) = \frac{1}{\|x_j - c_i\|_2} \tag{7}$$

where $I_i$ represents the superpoints within the $i$-th proposal and $x_j$ indicates the original coordinates of superpoints. Note that $*$ indicates the Hadamard product, which outputs the element-wise production of two vectors. After obtaining the coordinate $c_i$ and embedding $\boldsymbol{f}'_i$ for the $i$-th proposal, we then link the superpoints to proposal centroid for constructing the $k$-NN graph. To extract discriminative embedding of the proposal, we develop bilateral graph attention to achieve this. The bilateral weight $\boldsymbol{w}_{i,j}$ between the superpoint $j \in I_i$ and the $i$-th proposal is formulated as:

$$\boldsymbol{w}_{i,j} = \phi(\boldsymbol{f}'_i, \boldsymbol{f}_j) * \varphi(c_i, x_j) \tag{8}$$

where $\phi(\cdot, \cdot) : \mathbb{R}^C \times \mathbb{R}^C \to \mathbb{R}^C$ and $\varphi(\cdot, \cdot) : \mathbb{R}^3 \times \mathbb{R}^3 \to \mathbb{R}^C$ are two mapping functions implemented by MLP networks. $\phi(\boldsymbol{f}'_i, \boldsymbol{f}_j) = \text{ReLU}(\boldsymbol{W}_\phi^\top (\boldsymbol{f}'_i - \boldsymbol{f}_j))$ encodes the difference between the superpoint and proposal centroid in the feature space, while $\varphi(c_i, x_j) = \text{ReLU}(\boldsymbol{W}_\varphi^\top (c_i - x_j))$ encodes the difference between the superpoint and proposal centroid in the coordinate space. Thus, $\boldsymbol{w}_{i,j} \in \mathbb{R}^C$ captures the channel-wise relationship between the superpoint and proposal in the coordinate and feature spaces. We use the softmax function to obtain normalized weight $\hat{\boldsymbol{w}}_{i,j}$ across the proposal $I_i$, which is written as:

$$\hat{\boldsymbol{w}}_{i,j} = \frac{\exp(\boldsymbol{w}_{i,j})}{\sum_{k \in I_i} \exp(\boldsymbol{w}_{i,k})} \tag{9}$$

Finally, we sum the weighted superpoint embeddings to obtain the proposal embedding, which is given by:

$$\hat{\boldsymbol{f}}_i = \sum_{j \in I_i} \boldsymbol{w}_{i,j} * \boldsymbol{f}_j \tag{10}$$

After obtaining the proposal embedding, we adopt a classification head and a score head to predict the class and score of the proposal $I_i$. In addition, we use a superpoint mask head to predict the superpoint score for masking the low-confidence superpoints within the proposal. Note that by using the superpoint mask head, we can generate the instance from the candidate proposal. According to these three heads, we use the cross-entropy as the classification loss $\mathcal{L}_{cls}$, the binary cross-entropy as the score loss $\mathcal{L}_{score}$, and the mean squared error as the mask loss $\mathcal{L}_{mask}$ to form the instance loss $\mathcal{L}_{ins} = \mathcal{L}_{cls} + \mathcal{L}_{score} + \mathcal{L}_{mask}$ for training the superpoint graph cut network.

### 3.3 Training and Inference

In the training process, the whole framework is optimized by a joint loss, which is defined as:

$$\mathcal{L}_{joint} = \mathcal{L}_{sem} + \mathcal{L}_{geo} + \mathcal{L}_{ins} \tag{11}$$

where $\mathcal{L}_{sem}$ is the conventional cross-entropy loss for semantic scores, $\mathcal{L}_{geo}$ is the geometry-aware edge loss for edge scores, and $\mathcal{L}_{ins}$ is the instance loss for instance classification, score prediction, and superpoint mask prediction. In the inference process, our method directly outputs instances after a forward pass of the network. Note that non-maximum suppression is not necessary for our method.

## 4 Experiments

### 4.1 Experimental Settings

**Datasets.** We conduct experiments on two benchmark datasets, ScanNet v2 [7] and S3DIS [1]. The ScanNet v2 dataset contains 1,613 3D scenes, which are split into 1,201 training, 312 validation, and 100 test scenes, respectively. The results of instance segmentation are evaluated on 18 object categories. We report the results on validation and hidden test set. The ablation study is conducted on the validation set. The S3DIS dataset has 272 3D scans in 6 different areas with 13 object classes. The instance segmentation is evaluated in all classes. We report Area 5 and 6-fold cross-validation results, respectively.

**Evaluation metrics.** Following the ScanNet v2 official protocol, we use the mean average precision as the evaluation metric for both ScanNet v2 and S3DIS. The mean average precision with IoU

Table 1: Instance segmentation results on the ScanNet v2 hidden test set. The reported results are from the ScanNet benchmark on 19/5/2022.

| Method | AP | bathtub | bed | bookshe | cabinet | chair | counter | curtain | desk | door | other | picture | fridge | s.curtain | sink | sofa | table | toilet | window |
|---|---|---|---|---|---|---|---|---|---|---|---|---|---|---|---|---|---|---|---|
| 3D-SIS [13] | 16.1 | 40.7 | 15.5 | 6.8 | 4.3 | 34.6 | 0.1 | 13.4 | 0.5 | 8.8 | 10.6 | 3.7 | 13.5 | 32.1 | 2.8 | 33.9 | 11.6 | 46.6 | 9.3 |
| GSPN [45] | 15.8 | 35.6 | 17.3 | 11.3 | 14.0 | 35.9 | 1.2 | 2.3 | 3.9 | 13.4 | 12.3 | 0.8 | 8.9 | 14.9 | 11.7 | 22.1 | 12.8 | 56.3 | 9.4 |
| 3D-MPA [8] | 35.5 | 45.7 | 48.4 | 29.9 | 27.7 | 59.1 | 4.7 | 33.2 | 21.2 | 21.7 | 27.8 | 19.3 | 41.3 | 41.0 | 19.5 | 57.4 | 35.2 | 84.9 | 21.3 |
| PointGroup [18] | 40.7 | 63.9 | 49.6 | 41.5 | 24.3 | 64.5 | 2.1 | 57.0 | 11.4 | 21.1 | 35.9 | 21.7 | 42.8 | 66.0 | 25.6 | 56.2 | 34.1 | 86.0 | 29.1 |
| SSTNet [25] | 50.6 | 73.8 | 54.9 | 49.7 | 31.6 | 69.3 | 17.8 | 37.7 | 19.8 | 33.0 | 46.3 | 57.6 | 51.5 | 85.7 | 49.4 | 63.7 | 45.7 | 94.3 | 29.0 |
| HAIS [3] | 45.7 | 70.4 | 56.1 | 45.7 | 36.4 | 67.3 | 4.6 | 54.7 | 19.4 | 30.8 | 42.6 | 28.8 | 45.4 | 71.1 | 26.2 | 56.3 | 43.4 | 88.9 | 34.4 |
| SoftGroup [37] | 50.4 | 66.7 | 57.9 | 37.2 | 38.1 | 69.4 | 7.2 | 67.7 | 30.3 | 38.7 | 53.1 | 31.9 | 58.2 | 75.4 | 31.8 | 64.3 | 49.2 | 90.7 | 38.8 |
| GraphCut (ours) | **55.2** | 100 | 61.1 | 43.8 | 39.2 | 71.4 | 13.9 | 59.8 | 32.7 | 38.9 | 51.0 | 59.8 | 42.7 | 75.4 | 46.3 | 76.1 | 58.8 | 90.3 | 32.9 |

thresholds of 50% and 25% are denoted as $AP_{50}$ and $AP_{25}$, respectively. Also, AP denotes the mean average precision with the IoU threshold from 50% to 95% with a step size of 5%. Additionally, following existing methods [41, 3, 37], we use mean coverage (mCov), mean weighted coverage (mWCov), mean precision (mPrec), and mean recall (mRec) for S3DIS evaluation.

**Implementation details.** Our model is trained on a single TITAN RTX GPU. We use the Adam optimizer with a base learning rate of 0.001 for the network training, which is scheduled by a cosine annealing. The voxel size is set to 0.02m. A graph-based segmentation method [9] and SSP+SPG [22, 21] are used to generate superpoints for ScanNet scene and S3DIS room, respectively. At training time, we limit the maximum number of points in a scene to 250k and crop the excess randomly. Due to the high point density of S3DIS, we randomly downsample its 1/4 points before cropping. At inference, the whole scene is fed into the network without downsampling and cropping. Note that we follow [3, 37] and use the statistical average instance radius of the specific class to refine the instances.

## 4.2 Benchmarking Results

**ScanNet v2.** We compare our model with recent state-of-the-art methods on the unreleased test set of ScanNet v2. Table 1 reports the results on the leaderboard of the official testing server . It can be observed that our method achieves the highest performance in terms of AP. The results on the leaderboard can demonstrate the effectiveness of our method for 3D instance segmentation.

Moreover, we evaluate our method on the validation set of ScanNet v2. From the results in Table 2, one can observe that the proposed GrapCut can achieve better results. In particular, our method brings 2.8% gains for the metric AP and 1.5% gains for the metric $AP_{50}$ to the second-best methods. In addition, we provide the visualization results of our GraphCut and SoftGroup [37] in Figure 3. We use the red rectangular boxes to show the differences between them. It can be observed that our method can generate good instances with clear boundaries for objects clustered together, such as chairs. SoftGroup relies on point grouping by using offset-shifted point coordinates, which cannot make full use of local geometric information of point clouds. Since our method can fully utilize the local geometric information of point clouds by constructing an edge score prediction network and a superpoint graph cut network, our method achieves better results than SoftGroup on these clustered objects.

Table 2: Instance segmentation results on the ScanNet v2 validation set.

| Method | AP | $AP_{50}$ | $AP_{25}$ |
|---|---|---|---|
| GSPN [45] | 19.3 | 37.8 | 53.4 |
| 3D-MPA [8] | 35.3 | 51.9 | 72.4 |
| PointGroup [18] | 34.8 | 56.9 | 71.3 |
| SSTNet [25] | 49.4 | 64.3 | 74.0 |
| HAIS [3] | 43.5 | 64.1 | 75.6 |
| SoftGroup [37] | 46.0 | 67.6 | 78.9 |
| GraphCut (ours) | **52.2** | **69.1** | **79.3** |

**S3DIS.** In Table 3, we list the results of Area 5 and 6-fold cross-validation on S3DIS. Regarding the evaluation of Area 5, our method can outperform all compared methods. It is worth noting that our model improves SoftGroup by 2.5% in terms of AP. For the 6-fold cross-validation of S3DIS, our method is superior to the state-of-the-art methods on most metrics.

http://kaldir.vc.in.tum.de/scannet_benchmark/semantic_instance_3d.php?metric=ap

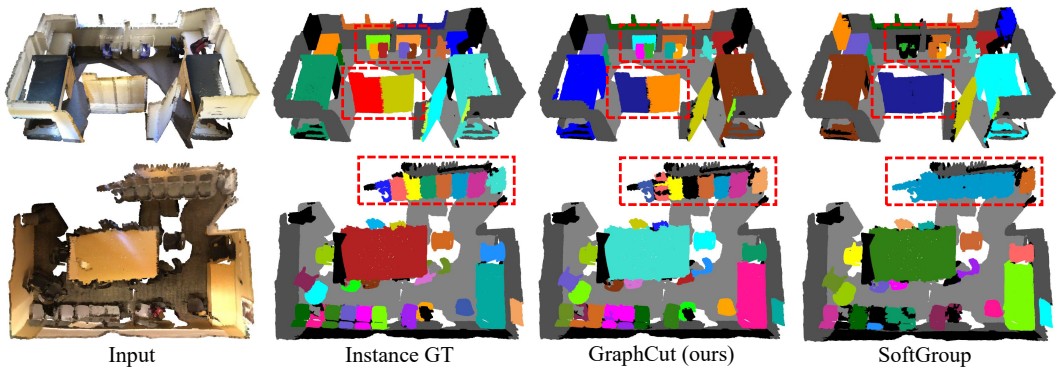

|  | Input | Instance GT | GraphCut (ours) | SoftGroup |

Figure 3: Visualization results of our GraphCut and SoftGroup [37] on the ScanNet v2 validation set.

Table 3: Instance segmentation results on the S3DIS dataset.

| Setting | Method | AP | $AP_{50}$ | mCov | mWCov | $mPrec_{50}$ | $mRec_{50}$ |
|---|---|---|---|---|---|---|---|
| Area 5 | SGPN [40] | - | - | 32.7 | 35.5 | 36.0 | 28.7 |
|  | ASIS [41] | - | - | 44.6 | 47.8 | 55.3 | 42.4 |
|  | PointGroup [18] | - | 57.8 | - | - | 61.9 | 62.1 |
|  | SSTNet [25] | 42.7 | 59.3 | - | - | 65.5 | 64.2 |
|  | HAIS [3] | - | - | 64.3 | 66.0 | 71.1 | 65.0 |
|  | SoftGroup [37] | 51.6 | 66.1 | 66.1 | 68.0 | 73.6 | 66.6 |
|  | GraphCut (ours) | **54.1** | **66.4** | **67.5** | **68.7** | **74.7** | **67.8** |
| 6-fold | SGPN [40] | - | - | 37.9 | 40.8 | 38.2 | 31.2 |
|  | PartNet [28] | - | - | - | - | 56.4 | 43.4 |
|  | ASIS [41] | - | - | 51.2 | 55.1 | 63.6 | 47.5 |
|  | 3D-BoNet [44] | - | - | - | - | 65.6 | 47.7 |
|  | OccuSeg [11] | - | - | - | - | 72.8 | 60.3 |
|  | GICN [26] | - | - | - | - | 68.5 | 50.8 |
|  | PointGroup [18] | - | 64.0 | - | - | 69.6 | 69.2 |
|  | SSTNet [25] | 54.1 | 67.8 | - | - | 73.5 | 73.4 |
|  | HAIS [3] | - | - | 67.0 | 70.4 | 73.2 | 69.4 |
|  | SoftGroup [37] | 54.4 | **68.9** | 69.3 | 71.7 | **75.3** | 69.8 |
|  | GraphCut (ours) | **56.3** | 68.2 | **72.8** | **75.0** | 74.4 | **73.7** |

## 4.3 Ablation Studies and Analysis

**Different $k$ in edge score prediction network.** In our edge score prediction network, we learn the similarity from the local $k$-NN graphs of two adjacent nodes to identify whether they belong to the same instance. Here, we study the impact of different $k$ on the instance segmentation performance. We select $k \in \{0, 2, 4, 8, 16\}$. Notably, $k$=0 means that we only concatenate two adjacent superpoint features as edge embedding. The results of AP, $AP_{50}$, and $AP_{25}$ are 52.0%, 68.8%, 78.7% ($k$=2), **52.2**%, **69.1**%, **79.3**% ($k$=4), 51.4%, 68.1%, 79.1% ($k$=8), and 50.8%, 67.7%, 79.1% ($k$=16), respectively. Since $k$=4 achieves the best results, we set $k$=4 in our experiment.

**Effectiveness of edge feature embedding.** To verify the effectiveness of our edge feature embedding, we consider three cases: (1) Only with edge embedding in coordinate space (dubbed as "Coordinate"), (2) Only with edge embedding in feature space (dubbed as "Feature"), (3) Only with embeddings of adjacent nodes as edge embedding. From the instance segmentation results on the ScanNet v2 validation set listed in Table 4, the best performance is achieved with the combination of the edge embeddings in both the coordinate and feature spaces. In the edge feature embedding, employing both geometry and feature embeddings of point clouds can improve the performance of the instance segmentation of point clouds.

Table 4: Ablation study on the ScanNet v2 validation set for edge feature embedding.

| Coordinate | Feature | AP | AP$_{50}$ | AP$_{25}$ |
|:---:|:---:|:---:|:---:|:---:|
| | | 50.1 | 66.6 | 77.4 |
| ✓ | | 50.9 | 68.0 | 78.3 |
| | ✓ | 51.5 | 68.8 | 78.5 |
| ✓ | ✓ | **52.2** | **69.1** | **79.3** |

Table 5: Ablation study on the ScanNet v2 validation set for geometry-aware edge loss.

| $\mathcal{L}_{area}$ | $\mathcal{L}_{feat}$ | AP | AP$_{50}$ | AP$_{25}$ |
|:---:|:---:|:---:|:---:|:---:|
| | | 47.8 | 65.1 | 75.3 |
| ✓ | | 51.7 | 68.6 | 78.8 |
| | ✓ | 51.2 | 67.8 | 77.8 |
| ✓ | ✓ | **52.2** | **69.1** | **79.3** |

**Ablation study on geometry-aware edge loss.** Here, we conduct the experiments on the ScanNet v2 validation set to verify the effectiveness of the propose geometry-aware edge loss. Specifically, we also consider three ablations: (1) Only with area constraint in the coordinate space (*i.e.*, "$\mathcal{L}_{area}$"), (2) Only with instance constraint in the feature space (*i.e.*, "$\mathcal{L}_{feat}$"), (3) Only with binary cross-entropy loss, *i.e.*, $\mathcal{L}_{edge}$. The results are listed in Table 5. It can be observed that the geometry constraints bring substantial gains to our method. By using the area constraint, it is easier to draw the nodes of the instance toward the instance center, making the boundary between different instances clearer.

**Effectiveness of bilateral graph attention.** In order to validate the effectiveness of the proposed bilateral graph attention, we replace the bilateral graph attention with a simple MLP network followed by max-pooling and conduct experiments on the ScanNet v2 validation set. The results of AP, AP$_{50}$, and AP$_{25}$ are 49.9%, 66.8%, 77.3% (MLP network), and **52.2%**, **69.1%**, **79.3%** (our bilateral graph attention), respectively. Without our designed bilateral graph attention, the performance drops a lot. This is because the bilateral graph attention can adaptively aggregate the information of superpoints in the same instance, which is more reasonable than the simple max-pooling operation for instance embedding.

## 5   Conclusion

In this paper, we proposed a learning-based superpoint graph cut method for 3D instance segmentation, which prunes the edges off the superpoint graph for generating instances. Specifically, we proposed an edge score prediction network with cross-graph attention in the coordinate and feature spaces to capture local geometric information of two adjacent nodes and predict the edge scores. A geometry-aware edge loss was proposed to train the edge score prediction network, which encourages two adjacent nodes in the same instance to be close to the instance center in both the coordinate and feature spaces. Based on the learned edge scores, a superpoint graph cut network was developed to cut irrelevant edges for instance generation. For the generated instances, we further adopted bilateral graph attention to predict semantic classes and scores of instances. Extensive experiments on ScanNet v2 and S3DIS benchmarks show that our method achieves new state-of-the-art performance on 3D instance segmentation.

## Acknowledgments

The authors would like to thank reviewers for their detailed comments and instructive suggestions. This work was supported by the National Science Fund of China (Grant Nos. U1713208, 61876084).

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
