# Supplementary Material for "Learning Superpoint Graph Cut for 3D Instance Segmentation"

**Le Hui[†], Linghua Tang[†], Yaqi Shen, Jin Xie[\*], Jian Yang[\*]**
PCA Lab, Nanjing University of Science and Technology, China
{le.hui, tanglinghua, syq, csjxie, csjyang}@njust.edu.cn

## A  Overview

This supplementary material provides more details on network architecture, visualization, and ablation study of our method. We also analyze the limitation and discuss the impact of our method. Specifically, in Sec. B, we provide specific network architecture and more details about the superpoint feature extraction. In Sec. C, we provide more visualization results, quantitative results, and ablation study of our proposed method. In Sec. D, we discuss the limitations and impacts of our method.

## B  Network Architecture

**Superpoint feature extraction.** To extract superpoint features, we first use the submanifold convolution to extract voxel-level features, where the raw 3D points are converted into voxels by performing voxelization. Specifically, we follow [5] and use a 3D U-Net structure constructed by five submanifold convolution blocks. The produced feature dimension of 3D U-Net is 32. Then, we average the voxel features belonging to the same instance to produce the initial superpoint features, where the superpoints are generated by using the method in [7]. After that, we adopt conditioned edge convolution network [6] on the superpoint graph to extract superpoint features, where the long-range context information can be captured through graph convolution. Finally, we can obtain 32-dimensional superpoint features. In addition, we use two classification heads based on voxel features and superpoint features to predict the semantic classes, respectively.

**Edge feature learning network.** In the edge feature learning network, we learn edge embeddings in both the coordinate and feature space to predict edge scores for proposing instances. For edge embeddings in the coordinate space, we first use a two-layer multi-layer perceptron (MLP) network to predict the 3-dimensional offsets of superpoints. Then, based on the shifted coordinate space, we construct the $k$-NN graphs for each node pair $(u, v) \in E$, where $E$ is the edge set on the superpoint graph. We adopt a three-layer MLP network to learn channel-wise attention weights in cross-graph attention for extracting edge embeddings. Similarly, for edge embeddings in the feature space, we also construct the $k$-NN graphs for each node pair $(u, v) \in E$, and adopt a three-layer MLP network to learn channel-wise attention weights in cross-graph attention for extracting edge embeddings. Finally, for each $(u, v) \in E$, we combine the edge embeddings in both the coordinate and feature spaces and the geometric distance ($L_2$ distance) between $u$ and $v$ in the shifted coordinated space. The two-layer MLP network followed by Sigmoid is used to produce edge scores of the superpoint graph.

**Superpoint graph cut network.** In the superpoint graph network, we present the bilateral graph attention to extract instance embeddings. Specifically, we first adopt the three-layer MLP network

---

[†]Equal Contributions, [\*]Corresponding authors.

Le Hui, Linghua Tang, Yaqi Shen, Jin Xie, and Jian Yang are with PCA Lab, Key Lab of Intelligent Perception and Systems for High-Dimensional Information of Ministry of Education, and Jiangsu Key Lab of Image and Video Understanding for Social Security, School of Computer Science and Engineering, Nanjing University of Science and Technology, China.

36th Conference on Neural Information Processing Systems (NeurIPS 2022).

on the coordinate and feature spaces to capture the geometry differences between the superpoints of the instance and the instance centroid, respectively. Here we can obtain two attention weights from the coordinate and feature spaces. Then, we execute element-wise production of the two attention weights to obtain the bilateral weight. After that, we use the softmax function to normalize the bilateral weight. Finally, we sum the weighed superpoint embeddings within the instance to obtain the instance embedding.

## C  More Results

### C.1  Quantitative Results

**ScanNet v2 test set.** In Table 1, we also report the 3D instance segmentation results on the ScanNet v2 test set in terms of AP, $AP_{50}$, $AP_{25}$. Note that the results in the table are mean average precision over 18 categories. It can be observed that our method achieves the best results in terms of AP and the second-best results in terms of $AP_{50}$ and $AP_{25}$. For $AP_{25}$, the results of our method are 14%, 12%, and 16% lower than those of SoftGroup [9] in the categories of the bookshelf, other furniture, and refrigerator, respectively. These three categories bring about 3% performance drop on the mean $AP_{25}$ over all categories. Similarly, these categories also lead to the performance drop in terms of $AP_{50}$. In addition, the samples of these three categories are much smaller than other categories, such as the chair, table, and cabinet. We visualize the ScanNet v2 test set and observe that the numbers of the bookshelf and refrigerator are about 12 and 13 samples. Due to the small number of samples, the performance on these categories fluctuates greatly. If you predict one more correct instance, the performance will be improved by about 8%. Therefore, about two object prediction errors cause the performance gap in these categories that have a small number of samples. Nonetheless, our method can achieve high-quality object instances with higher IoU scores. Since other methods cannot effectively obtain high-quality instances, $i.e.$, IoU score > 0.5, our method achieves the best results on AP.

**ScanNet v2 validation set.** In addition to the mean AP, $AP_{50}$, and $AP_{25}$ in the main paper, we report the detailed performance of each category in Table 2. Note that the results of SoftGroup [9] are obtained by using the pretrained models provided by the official code. Note that the obtained results are slightly lower than those listed in the main paper of SoftGroup. It can be observed that the performance gap between our method and SoftGroup is small on the bookshelf and refrigerator categories. Compared with the test set of 100 scenes, the validation set has 312 scenes. The numbers of samples of the bookshelf and refrigerator are 77 and 57, respectively. Compared with the performance gap in the test set of these two categories, the performance gap in the validation set is reduced. Considering our method performs well in most of the categories, our method achieves the best results in terms of AP, $AP_{50}$, and $AP_{25}$.

### C.2  Ablation Study

**Analysis of area constraint.** We perform the ablation study on the ScanNet v2 validation set to demonstrate the effectiveness of the area constraint used for learning compact offset on the superpoint graph. Specifically, we compute the mean absolute error (dubbed "Offset MAE") that indicates the $L_1$ distance between the shifted superpoint centers and the instance centers. We also compute the standard deviation (dubbed "Offset SD") of the $L_2$ distance between the shifted superpoint centers and the instance centers. In Table 3, we report the Offset MAE, Offset SD, as well as the AP, $AP_{50}$, and $AP_{25}$. It can be observed that our method equipped with the area constraint (dubbed "GraphCut w/ area constraint") can achieve the best results. In Figure 1, we also visualize the shifted superpoint centers, which are computed by adding the original superpoint centers and the predicted superpoint offsets. Note that the nodes of the superpoint graph "GraphCut w/ area constraint" and "GraphCut w/o area constraint" are colored with the same color as the graph-level instance ground truth for a better view.

**Analysis of soft threshold $\theta$.** To mitigate semantic prediction errors, we use the soft threshold $\theta$ to associate the superpoints with multiple classes. Here, we adjust the $\theta$ from 0 to 1 to conduct ablation studies on the ScanNet v2 validation set. Table 4 lists the average precision for different values of the $\theta$. It can be observed that the performance is stable between 0.1 to 0.4 for $\theta$. According to the experimental results, we set $\theta = 0.2$ in this paper.

Table 1: Instance segmentation results on the ScanNet v2 hidden test set in terms of mAP, mAP$_{50}$, and mAP$_{25}$. Note that the best results are highlighted in **bold** and the second-best results are underlined. The reported results are from the ScanNet benchmark on 19/5/2022.

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

Table 3: Ablation study on the ScanNet v2 validation set for area constraint. "Offset MAE" denotes the mean absolute error (*i.e.*, $L_1$ distance) between the shifted superpoint centers and the instance centers, and "Offset SD" denotes the standard deviation of the $L_2$ distance between the shifted superpoint centers and the instance centers. The best results are highlighted in **bold**.

| Method | Offset MAE | Offset SD | AP | $AP_{50}$ | $AP_{25}$ |
|---|---|---|---|---|---|
| GraphCut w/o area constraint | 0.373 | 0.182 | 51.3 | 68.0 | 78.6 |
| GraphCut w/ area constraint | **0.319** | **0.115** | **52.2** | **69.1** | **79.3** |

Table 4: Ablation study on the ScanNet v2 validation set for different values of the hyper-parameter $\theta$. The best results are highlighted in **bold**.

| Metrics | $\theta = 0.01$ | $\theta = 0.1$ | $\theta = 0.2$ | $\theta = 0.3$ | $\theta = 0.4$ | $\theta = 0.5$ | $\theta = 0.6$ | $\theta = 0.7$ | $\theta = 0.8$ | $\theta = 0.9$ |
|---|---|---|---|---|---|---|---|---|---|---|
| AP | 46.4 | 52.1 | **52.2** | 52.0 | 51.5 | 50.6 | 49.3 | 46.3 | 42.5 | 34.3 |
| $AP_{50}$ | 60.5 | 69.0 | 69.1 | **69.2** | 68.9 | 67.5 | 66.1 | 63.1 | 58.9 | 49.4 |
| $AP_{25}$ | 68.4 | 68.9 | **79.3** | 79.1 | 79.0 | 78.3 | 77.3 | 75.1 | 72.0 | 62.6 |

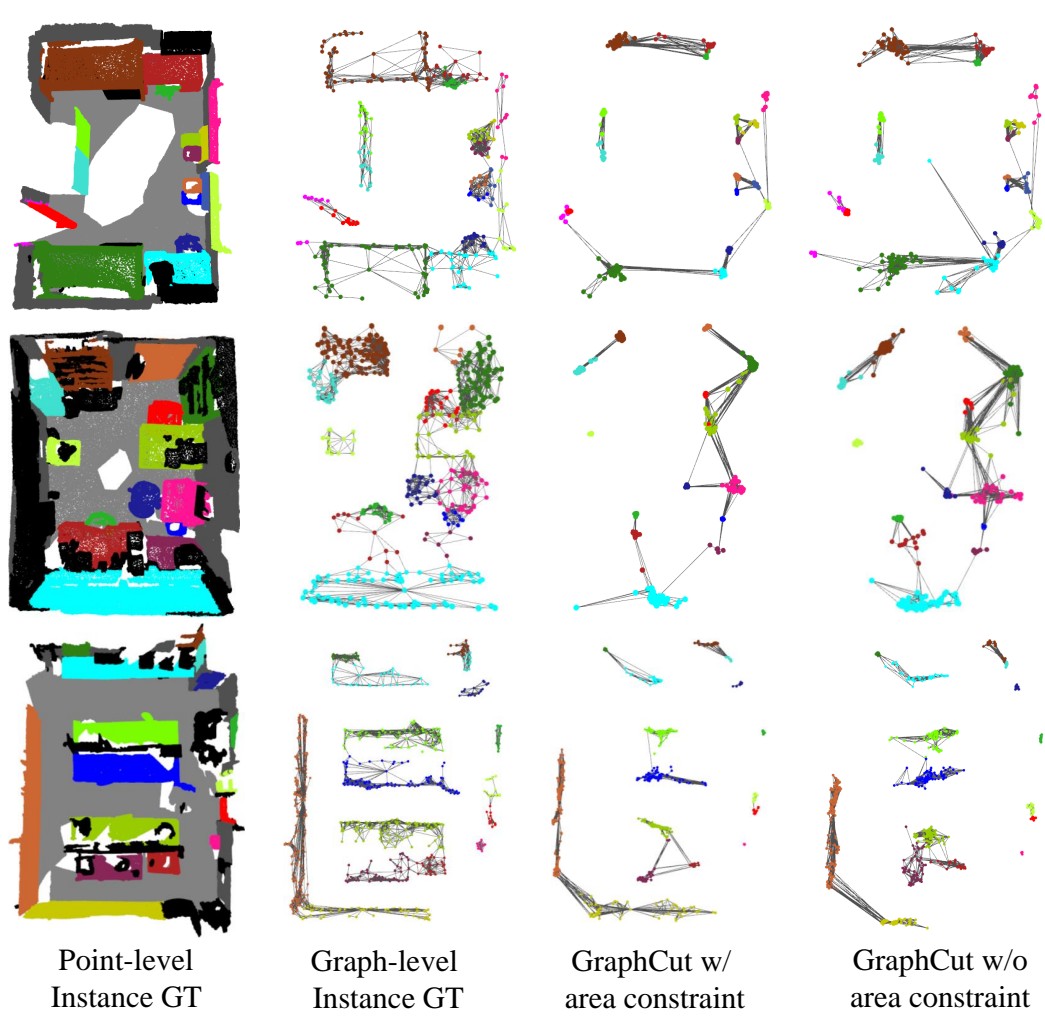

| Point-level Instance GT | Graph-level Instance GT | GraphCut w/ area constraint | GraphCut w/o area constraint |

Figure 1: The visualization results of superpoint offsets on the superpoint graph. Note that the nodes of the superpoint graphs "GraphCut w/ area constraint" and "GraphCut w/o area constraint" are colored with the same color as the graph-level instance ground truth for a better view.

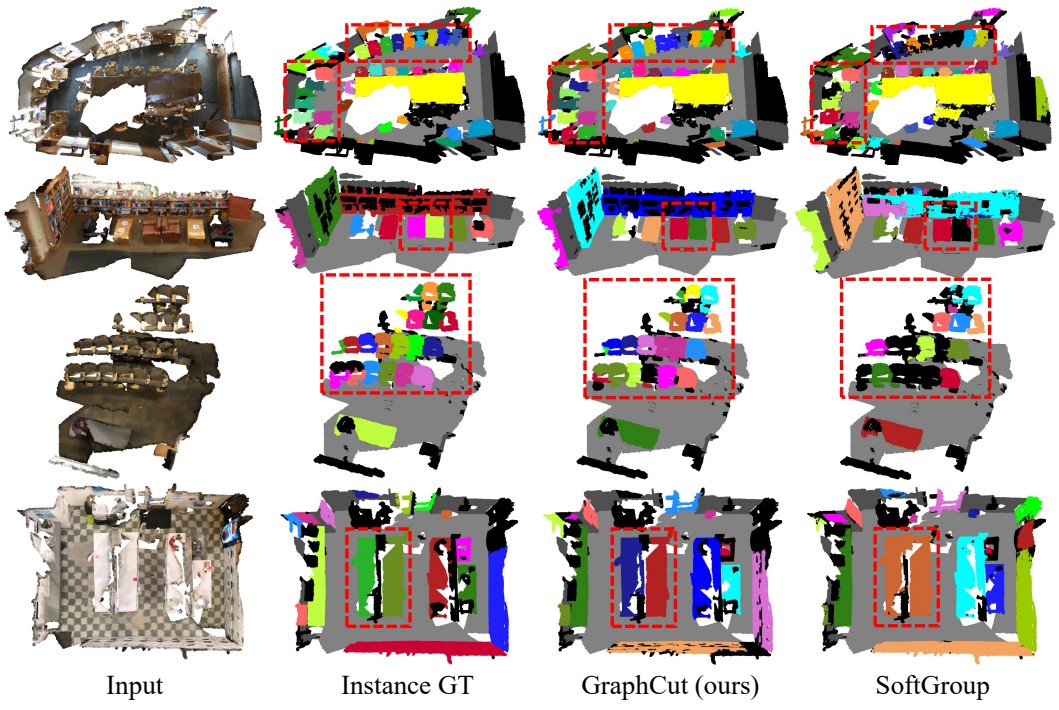

| Input | Instance GT | GraphCut (ours) | SoftGroup |

Figure 2: The visualization results of our method and previous state-of-the-art method SoftGroup [9] on the ScanNet v2 validation set. Red rectangles show the differences between the two methods of 3D instance segmentation.

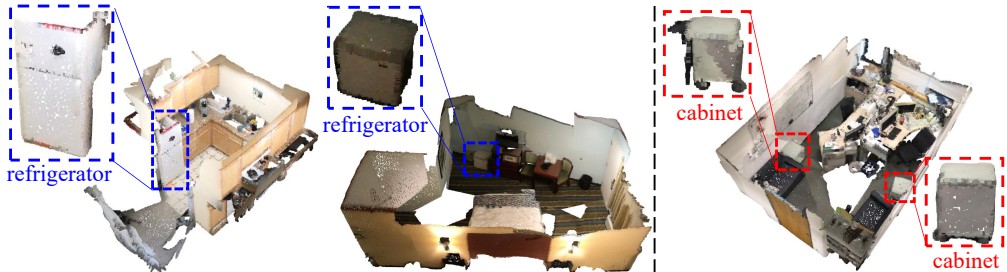

Figure 3: The visualization samples of the refrigerators and cabinets on the ScanNet v2 validation set.

## C.3 Visualization Results

**Visualization results**. In Figure 2, we show more visualization results of 3D instance segmentation on the ScanNet v2 validation set. Compared with SoftGroup [9], our method can effectively segment clustered objects, such as chairs.

**Visual process of graph cutting.** In order to show the detailed 3D instance segmentation process of our method, we provide the visualization results of each step of our method in Figures 4 and 5. Specifically, given a raw point cloud, we first oversegment it into superpoints and then construct superpoint graph. After that, we perform superpoint graph cutting on the superpoint graph, where the red edges are cut and the blue edges are remained. In this way, we can obtain graph-level instances of the point cloud. Finally, we convert graph-level instances into point-level instances. Note that the instances are randomly colored.

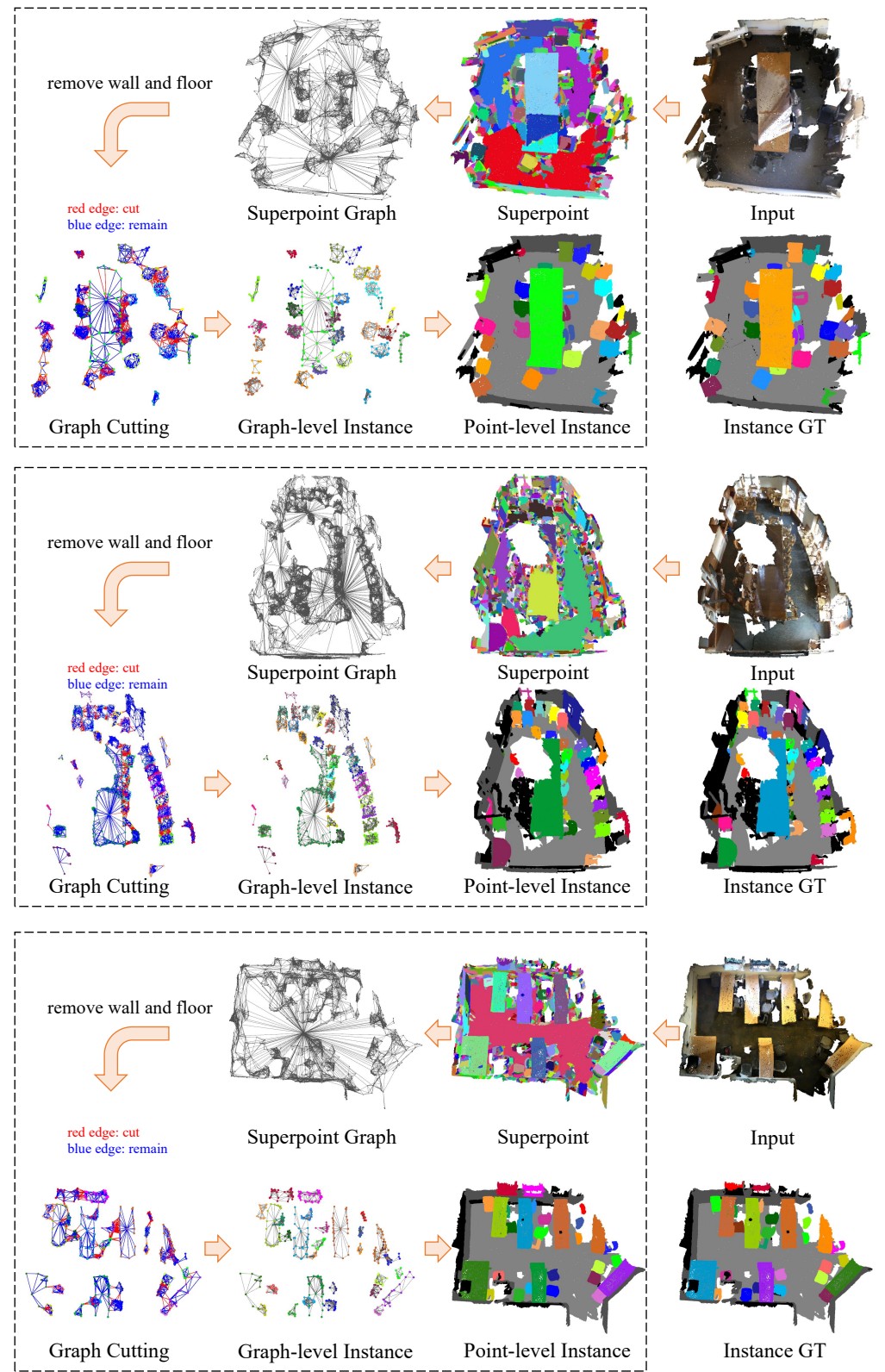

Figure 4: The 3D instance segmentation process of our method on the ScanNet v2 validation set. Note that instances are randomly colored.

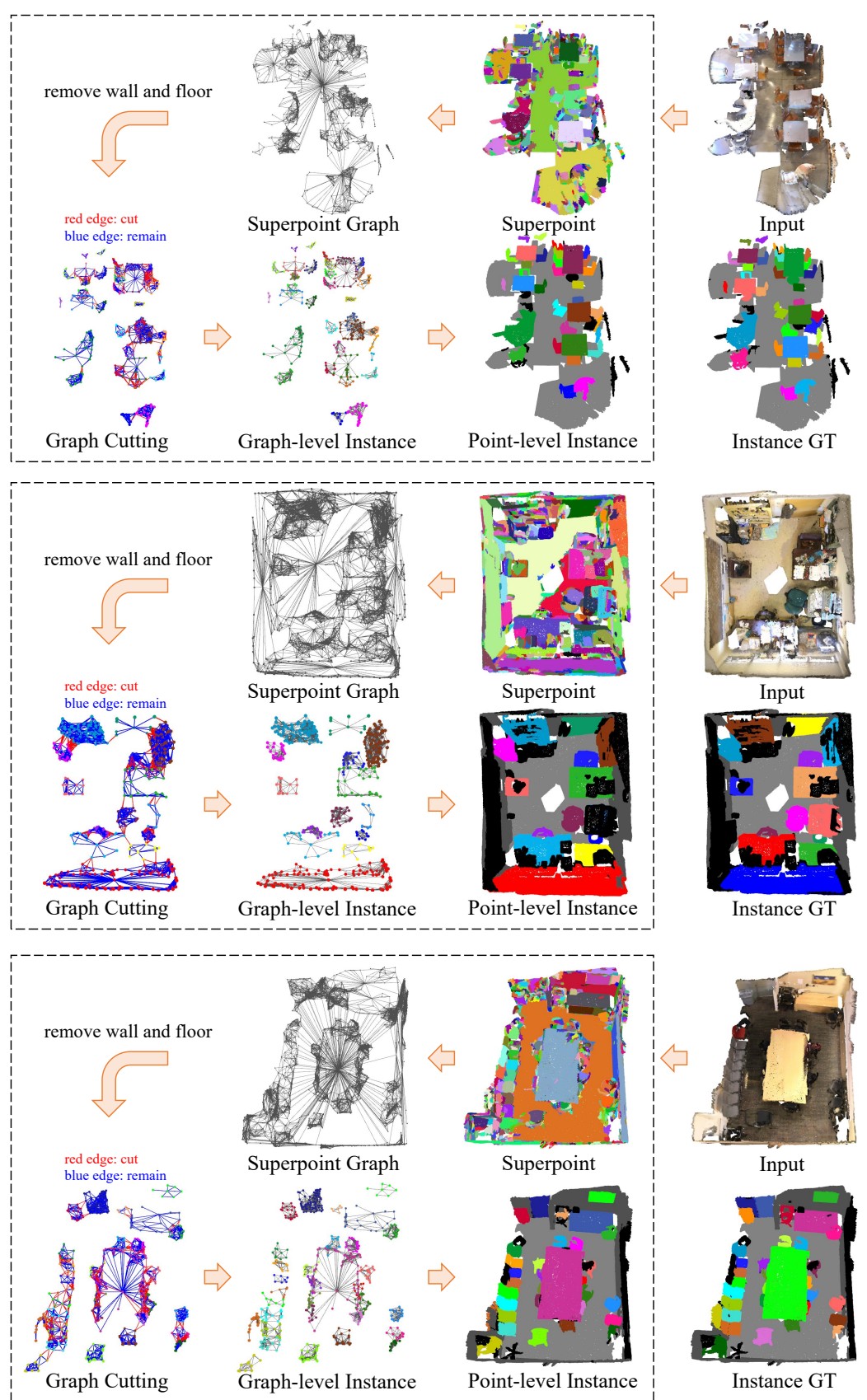

Figure 5: The 3D instance segmentation process of our method on the ScanNet v2 validation set. Note that instances are randomly colored.

Table 5: Inference time of different methods on the ScanNet v2 validation set. For a fair comparison, the runtime is computed on the same TITAN X GPU model.

| Method | Superpoint (ms) | Component Time (ms) | Total (ms) |
|---|---|---|---|
| SGPN [10] | - | Backbone (2080), Group merging (149000), Block merging (7119) | 158439 |
| ASIS [11] | - | Backbone (2083) Mean shift (172711), Block merging (7119) | 181913 |
| GSPN [13] | - | Backbone (2083), Point sampling (9559), Neighbour search (1500) | 12702 |
| 3D-BoNet [12] | - | Backbone (2083), SCN (667), Block merging (7119) | 9202 |
| GICN [8] | - | Backbone (1497), SCN (667), Block merging (7119) | 8615 |
| OccuSeg [3] | - | Backbone (189), Supervoxel (1202), Clustering (513) | 1904 |
| PointGroup [5] | - | Backbone (128), Clustering (221), ScoreNet (103) | 452 |
| SSTNet [7] | 195 | Backbone (125), Tree Network (229), ScoreNet (74) | 623 |
| HAIS [1] | - | Pointwise prediction (154), Hier.aggr. (118), Intra-inst.prediction (67) | 339 |
| SoftGroup [9] | - | Pointwise prediction (152), Soft grouping (123), Top-down refinement (70) | 345 |
| GraphCut (ours) | 195+15 | Extract superpoint features (122), Edge score prediction (5), Superpoint graph cut (42) | 379 |

## C.4 Inference Time

In Table 5, we report the average runtime of different methods on the ScanNet v2 validation set. Note that except for our method, the rest of the results in this table are derived from SoftGroup [9]. For a fair comparison, we use the same TITAN X GPU to evaluate the runtime of our method. In our GPU environment, we re-evaluated the runtime of SoftGroup and found that the runtime (343ms) is very close to the official time (345ms). Since SSTNet [7] and our GraphCut utilize the same method to generate superpoints, we add the runtime (195ms) of the superpoint generation to the total runtime. Note that our method also needs 15ms to construct the superpoint graph. For our GraphCut, we require 122ms for extracting superpoint features, 5ms for the edge score prediction network, and 42ms for the superpoint graph cut network. The total runtime of our method is 379ms (195+15+122+5+42). It can be observed that the runtime of our method outperforms most methods and is comparable to HAIS [1] and SoftGroup [9].

## D Limitations and Impacts

**Limitations.** In 3D instance segmentation, it requires to recognize the object instance and predict semantic categories simultaneously. According to the instance segmentation results on the ScanNet v2 validation set, we found that our method performs worse on the refrigerator category in terms of average precision (AP). Although our method can recognize the instances of the refrigerator, the semantic category of the instances is easily predicted to the cabinet, resulting in low AP. In Figure 3, we visualize the samples of refrigerators and cabinets in the ScanNet v2 validation set. It can be observed that mini refrigerators are very similar to cabinets, so mini-refrigerators can be easily classified as cabinets. In addition, we find that the number of samples of the refrigerator is much smaller than in other categories, such as chairs, tables, and desks. Therefore, the above two points lead to the low AP of the refrigerator category. In order to improve the AP of the refrigerator category, we can consider two points: (1) We can mine the context information of the refrigerator

to infer the refrigerator from the surrounding objects. (2) Executing data augmentation for those categories have a small number of samples.

**Impacts.** The proposed method has a potential impact in autonomous cars and transportation. For autonomous cars, object instances on the road may be incorrectly recognized by the proposed method, which will increase the risk of safe driving. These issues require further research and consideration when building upon this work for 3D instance segmentation in autonomous situation.

**Ethical consideration.** This work is able to facilitate the development of certain applications. For example, it can help domestic robots avoid potential obstacles in indoor environments. In assisted driving, it can help the driver recognize potential objects that may affect driving in advance. In addition, all datasets used in this paper are publicly available as academic research, and the evaluation metrics used in the experiments are also standard. For negative outcomes, it depends on a specific task and the criteria for assessing positive and negative.