# OpenReview forum: "Learning Superpoint Graph Cut for 3D Instance Segmentation"
_NeurIPS.cc/2022/Conference — NeurIPS 2022 Accept_

### Official Review · Reviewer_TEiW · 2022-07-10

**Rating:** 6
**Confidence:** 4
**Soundness:** 3 good
**Presentation:** 3 good
**Contribution:** 3 good

**Summary:**

This paper proposes a learnable superpoint-based graph cut method to explicitly learn the geometric structures for 3D point cloud instance problem.  Experiments on benchmark datasets show that the proposed method outperforms current state-of-the-art methods.

**Questions:**

Using attention to capture the similarities between points or nodes in graph is not a completely novel approach, there have already been some existing works [1] in 3D point cloud proposed before. What are the main differences of the author’s attention-based approach? It is suggested to append this mention in the Related Work section.

[1] Wang, Lei, et al. "Graph attention convolution for point cloud semantic segmentation." Proceedings of the IEEE/CVF conference on computer vision and pattern recognition. 2019.


**Limitations:**

What I’m more concerned about is the runtime efficiency on the proposed learnable graph cut method. For deploying the proposed framework in the autonomous driving application as mentioned in the Introduction section, a short response time is desired. From the supplemental, however, the runtime efficiency comparison only is performed on indoor datasets (S3DIS, ScanNet2). Outdoor dataset specific to autonomous driving such as SemanticKITTI should be the more suitable one for evaluation.

**Strengths And Weaknesses:**

Strengths:
The proposed approach which dynamically prunes the unrelated edges for better instance segmentation is relatively new. The paper presents a detailed description on the main building blocks (i) edge score prediction network and ii) superpoint graph cut network. And the ablation analysis validates the efficacy of the proposed network structure and geometry-aware edge loss.

Weaknesses:
Strictly speaking, the proposed approach is not completely learnable, since the initial superpoint graph oversegmented from point cloud is determined empirically. What is the sensitivity analysis on the different configurations on the initial superpoint graph?

---

> ### Author Response · Authors · 2022-08-02
> **Response to Reviewer TEiW**
>
> We thank the reviewer for the detailed comments to improve the paper.
>
> **Q1**: The proposed approach is not completely learnable, since the initial superpoint graph oversegmented from point cloud is determined empirically. The sensitivity analysis on the different configurations on the initial superpoint graph.
>
> **A1**: In this paper, we mainly focus on designing a learnable superpoint graph cut framework for 3D instance segmentation. In fact, different types of superpoint generation methods are flexible to use in our framework. Since the handcrafted superpoint generation methods are simple and efficient, we directly use the preprocessed handcrafted superpoints as the input. Moreover, we can integrate a learnable subnetwork (superpoint network for point cloud oversegmentation, Hui et al., ICCV 2021) into the current framework to generate superpoints, so as to form a completely learnable framework.
>
> In this paper, the used superpoint generation method is an optimization-based method, which first constructs a point-level graph based on the raw point clouds and then segments the edge to form the superpoints according to the edge energy computed by handcrafted geometric features. The critical threshold of edge energy affects the size and quality of superpoints. Note that the superpoint size indicates the number of points within it. In Area 5 of the S3DIS dataset, we study the effect of different superpoint sizes. By controlling the threshold in {-0.0005, -0.005, -0.05, -0.5, -5}, the generated average superpoint sizes in Area 5 are 356, 597, 725, 844, and 1080, respectively. We present the results as follows:
>
> | Average Superpoint Size | $AP$ | $AP_{50}$ | $AP_{25}$ |
> |:-:|:-:|:-:|:-:|
> | 356 | 52.6 | 64.1 | 69.8 |
> | 597 | 54.3 | 66.3 | **74.1** |
> | 725 | **54.4** | 66.1 | 74.0 |
> | 844 (used in this paper) | 54.1 | **66.4** | 73.6 |
> | 1080 | 52.4 | 65.2 | 72.5 |
>
> It can be observed that the performance is slightly floating for the superpoint sizes from about 350 to 1100. In this paper, for a fair comparison, we follow OccuSeg [8] and SSTNet [21] and use the same parameters to generate superpoints for ScanNet v2 and S3DIS datasets, respectively.
>
> &ensp;
>
> **Q2**: The main differences between the author’s attention-based approach and attention used in GACNet (Wang et al., CVPR 2019).
>
> **A2**: Although the formulas of our cross-graph attention and attention used in GACNet are similar, the differences are two folds: (1) GACNet learns the attention weights from a joint space by simply concatenating the coordinate and features space. However, our method learns the attention weights from two spaces, which are the shifted coordinate space and feature space. Since the features are learned from the original coordinate space, feature space does not match the shifted coordinate space (the learned offset vectors plus the original coordinates). (2) GACNet applies attention to the local k-NN graph of the coordinate space, while our method applies cross-attention to the non-local k-NN graphs in the shifted coordinate space and the feature space, respectively. By using the non-local k-NN graphs, our method can better characterize the similarities of adjacent nodes on the superpoint graph. In fact, in our edge score prediction network module, our main contribution focuses on learning dual similarities of superpoints from the shifted coordinate and feature spaces to produce accurate edge scores by using the proposed geometric-aware edge loss. It is flexible to use different attention mechanisms (such as self-attention and Point Transformer) to learn the similarity of adjacent nodes in the superpoint graph. In addition, we have cited the GACNet [33] in the related work of the main paper.
>
> &ensp;
>
> **Q3**: The runtime efficiency comparison only is performed on indoor datasets (S3DIS, ScanNet2). Outdoor dataset specific to autonomous driving such as SemanticKITTI should be the more suitable one for evaluation.
>
> **A3**: To the best of our knowledge, outdoor 3D datasets (such as SemanticKITTI, Waymo, and nuScenes) do not have the ground truth instance labels. Therefore, existing 3D instance segmentation methods are trained and evaluated on indoor datasets S3DIS and ScanNet v2. In order to estimate the runtime efficiency of our method on the outdoor datasets, we can roughly estimate the running time according to the number of points used for 3D instance segmentation. For example, the average number of points on the SemanticKITTI and ScanNet v2 is about 0.14 million and 0.12 million per room/scene, respectively. Since the numbers of points in SemanticKITTI and ScanNet v2 are at the same level, the running time of our method on SemanticKITTI may be similar to that on ScanNet v2 (running time 379ms reported in Table 4 of the supplementary material).

---

### Official Review · Reviewer_n9iC · 2022-07-11

**Rating:** 5
**Confidence:** 4
**Soundness:** 3 good
**Presentation:** 2 fair
**Contribution:** 2 fair

**Summary:**

This work tackles the task of 3D semantic instance segmentation by exploiting the local geometry. For that purpose, a graph is defined over geometric super-points. Then a series of graph cut networks, edge scoring functions and attention mechanisms is used to extract, classify and score the final instance mask. Experiments on S3DIS A5/6fCV and ScanNet test outperform prior published methods. Ablation studies on ScanNet validation show the effect of the individual model components.

**Questions:**

**Question**

The method introduces some new triangle losses, but it does not help? Table 3 in the supplementary seems to suggest that the L_area hurts the performance. However, Table 5 in the main paper suggests the opposite conclusion. What is the correct answer?

**Minor suggestions**

- l.133/149 - repeat meaning of F.
- Fig.1 - add the losses to the model figure (now it’s a bit unclear where they are applied), similarly add the used notation into the Figure -
- l.184 Parameter study for delta and beta?
- l.191 Where do the semantic classes come from?
- l.201 Parameter study?
- l.230 How is the scalar score obtained from the cross entropy loss / per-?
- l.235 Where is the semantic loss applied?
- l.278 Highlighting that the model produces clear boundaries is not that insightful - this is clear from the fact that the model is based on an over-segmentation which always produces clear boundaries.
- Training details: is the model trained in the train and val split for the test set submission?


**Limitations:**

Limitations and negative societal impact are adequately discussed in the rebuttal. In particular, the limitations include sensitivity to the long-tail problem, i.e, refrigerators which appear sparsely in the training data and expose a large intra-class variety. The papers also suggest potential solutions such as mining context information or data augmentation.

**Strengths And Weaknesses:**

**Strength**

The proposed approach is original in terms of the model architecture which is inspired by traditional graph cut approaches and achieves state-of-the-art 3D instance segmentation scores on two popular indoor scenes datasets (ScanNet, S3DIS).

**Weaknesses**

The model itself consists of numerous components that are not always very clearly explained and/or motivated. For example, l.122 mentions sparse convolutions, but it is not very clear if they are applied on points or on superpoints. Then additional edge-conditioned convolutions are applied. What is the motivation for two types of convolutions, wouldn’t one type of convolutions be enough? This could be further motivated and also evaluated to show that it actually improves performance. Overall, the model consists of numerous components which increases complexity and raises the question if a similar performance could be achieved with a simpler model, allowing the community to draw more general conclusions. At this stage, it is an interesting model that performs very well but I’m unsure about the significance to the community since it is unclear which conclusions to draw that can push the field forward.

The approach also relies on a large number of hyper-parameters (l.184, l.192, l.201, l.292 …), requiring manual tuning. The paper contains parameter studies only for a subset of them e.g. k in k-NN. From the paper it is unclear if the same hyperparameters are used for both datasets and how sensitive the approach is to the parameters.

---

> ### Author Response · Authors · 2022-08-04
> **Response to Reviewer n9iC (part 1)**
>
> We thank the reviewer for the detailed comments to improve the paper.
>
> **Q1**: It is not clear if sparse convolutions are applied on points or superpoints. Then additional edge-conditioned convolutions are applied. What is the motivation for two types of convolutions?
>
> **A1**: The sparse convolutions are applied on points to learn point features, while the edge-conditioned convolutions (i.e., graph convolution) are applied on the superpoint graph to learn superpoint features. Since the superpoint representation is coarser than the point representation, learning features directly from the superpoint representation cannot effectively capture the local geometric structures of point clouds. Thus, we first apply sparse convolutions on points to extract point-level features, and then use the point-level features to initialize superpoint-level features by averaging the point features of the same superpoint. Finally, based on the constructed superpoint graph, we apply edge-conditioned convolutions to extract superpoint features. In Sec. B of the supplementary material, we have provided the details on feature learning.
>
> &ensp;
>
> **Q2**: The model consists of numerous components which increases complexity and raises the question if a similar performance could be achieved with a simpler model, allowing the community to draw more general conclusions. It is an interesting model that performs very well but I’m unsure about the significance to the community since it is unclear which conclusions to draw that can push the field forward.
>
> **A2**: In 3D instance segmentation, existing superpoint-based methods (OccuSeg [8] and SSTNet [21]) directly use the superpoint features or construct the binary semantic tree to learn superpoint similarities, which are used to group instances. However, these methods cannot effectively capture the geometric context information of point clouds. To this end, we construct the superpoint graph to learn the geometric context similarities of superpoints and convert the instance segmentation into a binary classification of edges rather than grouping.
>
> Actually, our model mainly contains two core components: (1) The edge score prediction network applies cross-graph attention to the superpoint graph in double spaces (the shifted coordinate space and the feature space) for learning superpoint similarities, where the superpoint features are extracted from the sparse convolutions followed by the graph convolutions. (2) The superpoint graph cut network utilizes the BFS algorithm on the superpoint graph to generate proposals and applies bilateral graph attention to generate instances by using the learned edge scores. Compared to SSTNet, our simple yet efficient model can achieve state-of-the-art performance and shorter inference time.
>
> &ensp;
>
> **Q3**: The method introduces some new triangle losses, but it does not help? Table 3 in the supplementary seems to suggest that the L_area hurts the performance. However, Table 5 in the main paper suggests the opposite conclusion. What is the correct answer?
>
> **A3**: In Table 3 of the supplementary material, the items of “GraphCut w/ $L_{area}$” (the second row) and “GraphCut w/o $L_{area}$” (the third row) are typos. In fact, the second row should show the results with “GraphCut w/o $L_{area}$” and the third row should show the results with “GraphCut w/ $L_{area}$”, which is consistent with the conclusion of Table 5 of the main paper. Thank you for pointing out this problem. We will correct the typos in the revised manuscript.
>
> &ensp;
>
> **Q4**: Training details: is the model trained in the train and val split for the test set submission?
>
> **A4**: In this paper, the model trained in the training set and validation set is used for the online test set submission. According to the submission policy in the official document (https://kaldir.vc.in.tum.de/scannet_benchmark/documentation), existing methods follow this rule.
>
> &ensp;
>
> **Q5**: l.278 Highlighting that the model produces clear boundaries is not that insightful - this is clear from the fact that the model is based on an over-segmentation which always produces clear boundaries.
>
> **A5**: Although the oversegmented point clouds can produce superpoints with clear superpoint boundaries, the incorrectly grouped superpoints may lead to unclear instance boundaries. The quantitative and qualitative results (Tables 1, 2, and 3 and Figure 2 of the main paper) demonstrate that our method can generate high-quality instances with clear boundaries.
>
> &ensp;
>
> **Q6**: l.133/149 - repeat meaning of **F**.
>
> **A6**: The output **F** of the backbone network is used as the input for the two branches of the edge score prediction network. One branch uses **F** to learn superpoint similarities from the shifted coordinate space, and the other branch uses **F** to learn superpoint similarities from the feature space.

---

> > ### Author Response · Authors · 2022-08-04
> > **Response to Reviewer n9iC (part 2)**
> >
> > **Q7**: A large number of hyper-parameters, including $\delta$ and $\beta$ for triplet loss (on line 184) , soft threshold $\theta$ for semantic prediction (on line 192), threshold for edge cut (on line 201), and $k$ for cross $k$-NN graph (on line 292).
> >
> > **A7**: In 3D instance segmentation, the triplet loss (on line 184) is widely used to increase the distance between inter-instances and reduce the distance between intra-instances in the feature space. Edge cut (on line 201) is a binary classification problem. The edge will be cut from the superpoint graph if the edge score > 0.5, while the edge will be preserved if the edge score < 0.5. Thus, there are no hyper-parameters for edge cut. In fact, our method mainly contains two core hyper-parameters: a soft threshold $\theta$ for semantic prediction (on line 192) and the $k$ for cross $k$-NN graph (on line 292).
> >
> > (a) For the triplet loss (online 182), we follow OccuSeg [8] and set $\delta$ = 0.1 and $\beta$ = 1.5. We fix one of the hyper-parameters and adjust the other to perform ablation studies on the ScanNet v2 validation set. We present the average precision for different values of delta and beta as follows:
> >
> > | | $AP$ | $AP_{50}$ | $AP_{25}$ |
> > |:-:|:-:|:-:|:-:|
> > | $\delta$=0.1, $\beta$=0.5 | 49.8 | 67.1 | 76.7 |
> > | $\delta$=0.1, $\beta$=0.7 | 51.0 | 67.7 | 78.8 |
> > | $\delta$=0.1, $\beta$=0.9 | 51.2 | 68.4 | 78.3 |
> > | $\delta$=0.1, $\beta$=1.1 | 52.0 | 68.8 | 78.3 |
> > | $\delta$=0.1, $\beta$=1.3 | 51.9 | 68.9 | 79.2 |
> > | $\delta$=0.1, $\beta$=1.5 | **52.2** | 69.1 | 79.3 |
> > | $\delta$=0.1, $\beta$=1.7 | 52.0 | **69.2** | 79.3 |
> > | $\delta$=0.1, $\beta$=1.9 | 51.7 | 68.7 | 79.2 |
> > | $\delta$=0.3, $\beta$=1.5 | 52.0 | 69.0 | **79.4** |
> > | $\delta$=0.5, $\beta$=1.5 | 51.9 | 68.9 | 79.2 |
> > | $\delta$=0.7, $\beta$=1.5 | 51.6 | 68.9 | 78.3 |
> > | $\delta$=0.9, $\beta$=1.5 | 51.2 | 68.4 | 78.0 |
> > | $\delta$=1.1, $\beta$=1.5 | 50.8 | 67.9 | 77.9 |
> >
> > It can be observed that the performance is stable for delta and beta around at 0.1 and 1.5, respectively.
> >
> > (b) For the soft threshold theta (on line 192), we adjust the theta from 0 to 1 to perform ablation studies on the ScanNet v2 validation set. We present the average precision for different values of theta as follows:
> >
> > | | $AP$ | $AP_{50}$ | $AP_{25}$ |
> > |:-:|:-:|:-:|:-:|
> > | $\theta$=0.01 | 46.4 | 60.5 | 68.4 |
> > | $\theta$=0.1 | 52.1 | 69.0 | 68.9 |
> > | $\theta$=0.2 | **52.2** | 69.1 | **79.3** |
> > | $\theta$=0.3 | 52.0 | **69.2** | 79.1 |
> > | $\theta$=0.4 | 51.5 | 68.9 | 79.0 |
> > | $\theta$=0.5 | 50.6 | 67.5 | 78.3 |
> > | $\theta$=0.6 | 49.3 | 66.1 | 77.3 |
> > | $\theta$=0.7 | 46.3 | 63.1 | 75.1 |
> > | $\theta$=0.8 | 42.5 | 58.9 | 72.0 |
> > | $\theta$=0.9 | 34.3 | 49.4 | 62.6 |
> >
> > It can be observed that the performance is stable for $\theta$ from 0.1 to 0.4. We set $\theta$ = 0.2 in this paper. We will add the results of the ablation study in the revised manuscript.
> >
> > (3) For the hyper-parameter $k$ (on line 292), please refer to ablation studies in Sec 4.3 of the main paper.

---

### Official Review · Reviewer_nE6M · 2022-07-11

**Rating:** 7
**Confidence:** 4
**Soundness:** 3 good
**Presentation:** 3 good
**Contribution:** 3 good

**Summary:**

This is a good paper with new idea and solid experiments.

**Questions:**

1.Line 133, the authors design a MLP to predict the offset vectors. Is there any constrain on the MLP? Could you please add a figure (in the supp.) to show the shifted points or superpoints? Could you please add one experiment that without the offset vectors?
2.Line 172, for nodes u and v, if they are from different instances, the triangle areas are also  minimized, just as they are from the same instance. Please explain it in detail.

**Limitations:**

Please refer to the Questions.

**Strengths And Weaknesses:**

Strengths: 1. The idea is novel. 2. The experiments are sufficient. 3. The results are good.
Weaknesses: Refer to the Questions.

---

> ### Author Response · Authors · 2022-08-02
> **Response to Reviewer nE6M**
>
> We thank the reviewer for the comments to improve the quality of our paper.
>
> **Q1**: Is there any constraint on the MLP for predicting the offset vectors?
>
> **A1**: There are two constraints on the MLP for predicting the superpoint offset vectors. (1) In the area constraint loss ($L_{area}$), we minimize the $L_2$ distance between the shifted superpoint and the corresponding instance center, where the shifted superpoint is obtained by adding the superpoint coordinate with the predicted offset vector. Thus, the $L_2$ loss is explicitly imposed on the MLP. (2) Based on the shifted superpoints, we learn the similarity of superpoints in the shifted coordinate space for predicting edge score, which is supervised by a binary cross-entropy loss ($L_{edge}$). Thus, the binary cross-entropy loss is implicitly imposed on the MLP.
>
> &ensp;
>
> **Q2**: Add the figure of the shifted points or superpoints in the supplementary material.
>
> **A2**: In Figure 1 of the supplementary material, we visualize the shifted superpoints (the third column) on the superpoint graph. Compared with the original superpoint graph (the second column), the superpoints of the same instance approach to the corresponding instance centers by applying the learned offset vectors.
>
> &ensp;
>
> **Q3**: Add one experiment that without the offset vectors.
>
> **A3**: In Table 4 of the main paper, we show the results of the ablation study with or without offset vectors. For a clear comparison, we present the corresponding results as follows:
>
> | GraphCut | $AP$ | $AP_{50}$ | $AP_{25}$ |
> |:-:|:-:|:-:|:-:|
> | w/o offset vectors | 51.9 | 68.8 | 78.5 |
> | w/ offset vectors | **52.2** | **69.1** | **79.3** |
>
> It can be observed that without using the offset vectors, the performance will degrade. As offset vectors draw the superpoints toward the corresponding instance centers, the spatial distance between the superpoints of different instances is increased, which benefits the edge cut for generating high-quality instances.
>
> &ensp;
>
> **Q4**: For nodes $u$ and $v$, if they are from different instances, the triangle areas are also minimized, just as they are from the same instance. Please explain it in detail.
>
> **A4**: For nodes $u$ and $v$ from the same instance, we minimize the area of one triangle to reduce the distance between them. For nodes $u$ and $v$ from different instances, we minimize the areas of two triangles to increase the distance between them. To achieve this, we introduce the area constraint ($L_{area}$) in Eq.(4) of the main paper. In fact, $L_{area}$ contains two terms: the $L_2$ distance between the node and its corresponding instance center and the areas of the constructed triangles. Ideally, the optimal solution of $L_{area}$ is obtained when $u$ and $v$ shift to the corresponding instance centers. With the iterative optimization of the network training, $u$ and $v$ are gradually close to the corresponding instance centers. As shown in Figure 1 of the supplementary material, for nodes $u$ and $v$ from the same instance, $u$ and $v$ are simultaneously close to the common instance center. Thus, they are pulled close to each other in the coordinate space, which is helpful to group $u$ and $v$ into the same instance. For $u$ and $v$ from different instances, $u$ and $v$ are respectively close to the corresponding instance centers. Thus, they are pushed away in the coordinate space, which is helpful to divide $u$ and $v$ into two different instances. In Sec C.2 of the supplementary material, we have conducted experiments to verify the effectiveness of minimizing the area of the triangles. Quantitative results (Table 3 of the supplementary material) show that our method can obtain better results by minimizing the triangle areas.

---

### Meta-Review · Area_Chair_YXpG · 2022-08-27

**Recommendation:** Accept
**Confidence:** Certain

**Metareview:**

The paper follows the recent trend of segmenting point clouds into instances using superpoints. Technically, the paper proposes to construct superpoint graph, and learns to perform graph cut for the segmentation. Three reviewers generally appreciate the idea and empirical results, although some architectural components and hyperparameters are less evaluated.

After the rebuttal phase, three reviewers hold their original decision on accepting the paper. AC agrees.  Congratulations!


**Award:**

No

---

### Decision · Program_Chairs · 2022-09-14

Accept